# Quantitative Evaluation of Image Quality and Radiation Dose Using Novel Intelligent Noise Reduction (INR) Software in Chest Radiography: A Phantom Study

**DOI:** 10.3390/diagnostics15111391

**Published:** 2025-05-30

**Authors:** Ahmed Jibril Abdi, Helle Precht, Claus Bjørn Outzen, Janni Jensen

**Affiliations:** 1Department of Clinical Engineering, Region of Southern Denmark, Odense University Hospital, J.B. Winsløws Vej 4, 5000 Odense, Denmark; 2Research and Innovation Unit of Radiology, University of Southern Denmark, Kløvervænget 10, 5000 Odense, Denmark; janni.jensen@rsyd.dk; 3Department of Regional Health Research, University of Southern Denmark, Campusvej 55, 5230 Odense, Denmark; hepr@ucl.dk; 4Health Sciences Research Centre, UCL University College, Niels Bohrs Allé 1, 5230 Odense, Denmark; 5Department of Radiology, Lillebaelt Hospital, University Hospitals of Southern Denmark, Sygehusvej 24, 6000 Kolding, Denmark; 6Education of Radiography, UCL University College, Niels Bohrs Allé 1, 5230 Odense, Denmark; clbo@ucl.dk; 7Department of Radiology, Odense University Hospital, J. B. Winsløws Vej 4, 5000 Odense, Denmark; 8CAI-X (Centre for Clinical Artificial Intelligence), Odense University Hospital, University of Southern Denmark, 5000 Odense, Denmark

**Keywords:** intelligent noise reduction, quantitative image quality optimisation, chest protocol, radiation exposure to the patients, virtual anti-scatter grid

## Abstract

**Background/Objectives:** This study quantitatively evaluates the novel Intelligent Noise Reduction (INR) software NE 3.10.0.15 across three chest radiography protocols, namely, physical anti-scatter grid, non-grid, and virtual anti-scatter grid, to optimise the patient radiation dose while maintaining sufficient image quality. **Methods:** Quantitative image quality and radiation dose were evaluated using a CDRAD phantom with 20 cm PMMA to simulate the patient across three chest protocol settings at INR levels of 0, 5, and 8 for both PA and LAT projections. Effective doses were estimated using PCXMC Monte Carlo simulation software 2.0. **Results:** The findings revealed significant improvements in image quality with increasing INR levels, with INR8 consistently outperforming INR5 and non-INR settings. Protocols employing virtual or no grid achieved substantial radiation dose reductions of 77–82% compared to the physical grid. The virtual grid did enhance the quantitative image quality by 6–9% compared to non-grid configurations. **Conclusions:** INR software, particularly when combined with virtual anti-scatter grids, offers a promising solution for improving image quality while significantly reducing the patient radiation dose in chest radiography. Future clinical validation, incorporating subjective visual assessments by radiologists, is recommended to confirm these findings and facilitate the integration of INR closer to clinical practice.

## 1. Introduction

Optimising image quality while minimising radiation exposure is crucial in digital radiography [1,2]. Achieving this balance is particularly important, as diagnostic accuracy must be maintained without subjecting patients to an unnecessary dose.

In recent years, there has been increasing interest in integrating artificial intelligence (AI) into medical imaging systems to improve image quality and optimise radiation exposure, ultimately benefiting patients [3]. Noise is a significant factor that can affect the quality of clinical radiographic images, arising from various sources, including scattered radiation, image detector components, and reduced exposure parameters [3]. Integrating noise reduction in clinical protocols of the X-ray systems is, therefore, a promising approach to improving image quality and potentially providing a more accurate diagnosis for patients [4,5,6]. Recently, a promising AI-based software algorithm for noise reduction has been developed and integrated into the digital radiography (DR) X-ray system, called Intelligent Noise Reduction (INR).

According to the technical and clinical image quality and radiation assessment white papers, this AI-based INR software presents a promising solution that reportedly combines improved image quality through enhanced contrast and the effective removal of noise artefacts with reduced radiation exposure for patients [7,8]. It also has the potential to enhance the visualisation of different anatomical structures, thereby supporting the production of diagnostically accurate images at lower doses [9,10]. The potential benefits of INR software are particularly significant for radiation-sensitive patients, such as paediatric populations, who are more vulnerable to the adverse effects of ionising radiation [2]. A clinical study conducted at Dayton Children’s Hospital demonstrated that the use of INR software enabled a 50% reduction in radiation dose while maintaining diagnostic image quality [8]. The findings indicated that the software effectively reduced image noise at lower dose levels, supporting its clinical applicability in dose-sensitive patient groups [8]. This is consistent with the radiation protection principle of keeping exposure “as low as reasonably achievable” (ALARA) [11,12].

The use of a physical anti-scatter grid in digital radiography offers both advantages and disadvantages. While it effectively reduces scattered radiation and enhances image quality, it also contributes to an increased radiation dose to the patient [5]. In contrast, the use of a virtual anti-scatter grid has been shown to reduce patient radiation exposure without compromising image quality [13,14,15]. The combination of virtual anti-scatter grid technology with INR software is therefore expected to have a complementary effect, enhancing image quality while maintaining low radiation dose, an aspect evaluated in this study.

The aim of this study was to quantitatively evaluate the impact of INR software on image quality and radiation dose across three chest radiography protocol settings: employing a physical anti-scatter grid, no grid, and a virtual anti-scatter grid (software-based scatter correction), including three levels or combinations of INR. The underlying hypothesis was that INR would reduce image noise, thereby enhancing image quality while supporting radiation dose optimisation. It investigates how varying levels of INR software influence these parameters, offering valuable insights into the potential benefits and limitations of the technology in optimising radiographic image quality while minimising radiation dose.

## 2. Materials and Methods

### 2.1. Imaging and Image Acquisition Systems

The Arcoma i5 digital radiography system (Arcoma AB, Växjö, Sweden), equipped with the Canon image acquisition platform (CXDI Control Software NE, CPI CMP200DR) and a Canon wall stand flat panel detector (CXDI-401C Compact, Canon Europa, Amsterdam, The Netherlands), was used in this study.

Image quality and patient radiation exposure were assessed using INR AI-based noise reduction software, NE 3.10.0.15 (Canon Europe, Amsterdam, The Netherlands).

Scatter correction software NE V2.14, integrated within the Canon platform and functioning as a virtual anti-scatter grid, was also used.

The Arcoma i5 system was used, equipped with a physical anti-scatter grid (grid focus 180 cm and grid ratio 10:1) to reduce scatter and enhance image contrast.

### 2.2. Imaging Quality Evaluation Phantoms

A CDRAD 2.0 phantom from Artinis Medical Systems B.V. (Elst, The Netherlands) was used to evaluate the quantitative image quality at different chest protocol settings and INR level settings. The CDRAD 2.0 phantom is a device employed to evaluate the imaging performance of radiological systems. It comprises a polymethyl methacrylate (Plexiglas) tablet incorporating 225 squares, arrayed in a 15 × 15 matrix. Each square contains either one or two cylindrical bores of precisely defined diameter and depth, manufactured to tolerances of ±0.02 mm. The initial three rows of squares each contain a single bore, whilst the subsequent rows each feature two identical bores—one centrally located and the second positioned randomly within a corner. This design facilitates a thorough assessment of image quality parameters [2].

A 40 × 40 cm PMMA block with a thickness of 20 cm was used in combination with a 1 cm thick CDRAD 2.0 phantom, resulting in a total attenuation equivalent to 21 cm of PMMA [16]. This configuration closely simulates the anterior–posterior dimension of an average adult chest, corresponding to a body weight of approximately 70–75 kg [17,18]. The configuration also aligns with standard practice in diagnostic radiography and is commonly used for quality assurance, dosimetry, and equipment calibration [19].

### 2.3. Dosimeters

A Piranha 657 solid-state dosimeter (RTI Group in Mölndal, Sweden) was used to measure air kerma (AK) and validate the dose area product (DAP) meters of the radiography system. In addition, annual routine quality control procedures were performed for the system’s DAP meter. The Piranha 657 dosimeter, which is routinely calibrated, is equipped with a backscatter shield designed to eliminate scattered radiation, enhancing the precision of radiation dose measurements [20].

### 2.4. Software

The widely used Patient-Centred X-ray Monte Carlo (PCXMC) 2.0 software developed by the Finnish Radiation and Nuclear Safety Authority (STUK) was used to simulate and calculate the effective dose (ED) for patients.

Statistical Package for the Social Sciences (SPSS, Release 26.0.0.0, New York, NY, USA) was applied for all statistical analyses and statistical comparisons of the image quality and radiation dose to the patients of different protocol settings.

The CDRAD Analyser 2.1 software program (Artinis Medical Systems, Elst, and The Netherlands) was used to analyse radiographic images of the CDRAD 2.0 contrast–detail phantom.

### 2.5. Quantitative Image Quality Assessment

Contrast–detail resolution in X-ray imaging refers to the ability of the imaging system to display and distinguish fine details and subtle differences in contrast. It is an important measure of the system’s ability to depict small structures or differences in tissue density within an image. Contrast–detail resolution is typically evaluated using a phantom or test object specifically designed for this purpose, such as the CDRAD test object. Through the acquisition and analysis of X-ray images of this phantom, contrast resolution can be quantitatively and objectively assessed.

The evaluation of contrast–detail resolution involved assessing the visibility and discriminability of specific patterns or objects across varying contrast levels and spatial frequencies. To optimise this resolution, imaging parameters were systematically adjusted to improve the visibility of fine anatomical structures and subtle abnormalities in the X-ray images. This approach was intended to support more accurate detection and interpretation by radiologists, ultimately contributing to enhanced diagnostic precision and reliability [21].

Figure 1 illustrates the setup used to measure the contrast–detail resolution for all chest protocol settings. To minimise scattered radiation reaching the PMMA plates, the CDRAD test object was positioned between PMMA attenuation blocks, both precisely centred within the X-ray beam. Ten images of the phantom were acquired for each protocol setting to allow a comprehensive evaluation.

To ensure an objective evaluation of the image quality of the chest protocol settings, the CDRAD Analyser 2.0 software package was used. This software automatically evaluated the acquired images of the CDRAD test object and thus enabled a comprehensive analysis.

Ten input images acquired with uniform radiological factors were required to generate a contrast–detail curve. The CDRAD phantom images in DICOM format were retrieved directly from the imaging modality for analysis.

As shown in Figure 1, the X-ray tube generates the X-ray beam, which passes through the PMMA plates and the CDRAD test object before being detected by the image detector. The PMMA plates serve as attenuation material, and the CDRAD test object is positioned in the centre of the beam to evaluate contrast–detail resolution. The source-to-image distance (SID) is indicated, ensuring proper alignment and geometry for the X-ray acquisition.

The CDRAD analysis software computes the inverse image quality figure (IQF_inv_), a phantom-based image quality metric that integrates object contrast and diameter to quantify contrast–detail detectability. In addition, the software generates a contrast–detail curve (CDC) and calculates the percentage of correctly identified holes (out of 166), enabling an objective evaluation of detection accuracy [16,22,23]. Compared to conventional pixel-based metrics such as the signal-to-noise ratio (SNR) and contrast-to-noise ratio (CNR), IQF_inv_ provides a more comprehensive assessment of image quality, particularly in evaluating low-contrast resolution performance.
(1)$$\mathrm{Correct}\;\mathrm{observation}\;\mathrm{ratio}=\frac{\mathrm{Correct}\;\mathrm{observations}}{\mathrm{Total}\;\mathrm{number}\;\mathrm{of}\;\mathrm{squares}}\cdot 100\%$$

The CDRAD analyser calculates the image quality figure (IQF), a key metric quantifying contrast–detail visibility in CDRAD phantom radiographic images. Lower IQF values indicate better contrast resolution and are determined using Equation (1) [24,25].
(2)$$IQF=\sum_{i=1}^{15}C_{i}\cdot D_{i,j}$$
where D_i,j_ denotes the threshold (j) diameter in contrast to columns i and C_i_ the correctly identified contrast values.

The evaluation of radiographic image quality often involves the use of the IQF_inv_. This parameter, as defined by Equation (2) [24,26], serves as an indicator of image quality, with higher IQF_inv_ values corresponding to better image quality.
(3)$${IQF}_{inv}=\frac{100}{\sum_{i=1}^{15}C_{i}\cdot D_{i,j}}$$

Figure 2 provides a representative example of the CDRAD image quality phantom, including both a photographic image and the corresponding radiographic image, to illustrate the experimental setup and contrast–detail evaluation.

### 2.6. Radiation Dose Measurement

Air kerma (AK) was measured for each protocol to verify the integrated DAP meter in the imaging system.

The AK in radiography is usually measured in milligrays (mGy), which represents the absorbed radiation dose.

The DAP is a radiation dose quantity that reflects the total amount of radiation delivered to a specific area during a radiographic procedure. It combines the AK measured at the patient’s entrance surface with the size of the irradiated field. DAP is also used to calculate the ED, which is the factor used to estimate the stochastic risk of radiation-induced cancer [27].

To verify the imaging system DAP, 20 cm thick PMMA attenuation plates were positioned on the image detector to simulate the attenuation and scatter typical of average adult chest patients.

A Piranha solid-state dosimeter, shielded against backscatter, was positioned on the PMMA blocks to measure the input exposure of the patients. The dosimeter allowed direct measurement of the input exposure (air kerma) in each protocol setting of the imaging system.

### 2.7. Effective Dose Estimation

The ED is a calculated value representing the overall radiation dose received by a person, taking into account the different sensitivities of different organs and tissues to radiation. It is estimated by combining individual organ doses weighted by radiation sensitivity factors as specified in international radiation protection guidelines [28,29]. In this study, the ED received by the patient was estimated using the validated values for the DAP and by simulating the dose with the Monte Carlo software programme PCXMC 2.0. The ED was calculated according to the methodology described in publication 103 of the International Commission on Radiological Protection (ICRP, 2007) [29].

### 2.8. Clinical Examination Protocols

To assess the impact of INR software on image quality and patient radiation exposure, a clinical chest X-ray protocol was used. This protocol was chosen for its clinical significance and frequent use in radiology [30]. As these protocols are fundamental to diagnosing thoracic conditions, as evaluating the INR software’s performance prior to clinical implementation is vital. The evaluation compared different levels of INR software application and various chest X-ray protocol settings with the conventional protocol. Specifically, it examined a protocol with a physical anti-scatter grid at a higher dose level, one without a physical anti-scatter grid, and one using a virtual anti-scatter grid (scatter correction software NE.V2.14).

To determine the optimal image quality with the most radiation dose-saving protocol configuration of the INR software, different levels of INR software and protocols with varying radiation doses were compared to the default protocol setting. In addition, the inclusion of the virtual anti-scatter grid evaluates the combined effect of noise and scatter reduction on image quality and radiation dose. The following protocol settings listed in Table 1 are used to evaluate the software considering image quality and radiation dose to patients.

### 2.9. Statistical Analysis

Since the distribution of the quantitative image quality metrics (IQF_inv_) for the different chest protocol settings was not confirmed, the Kruskal–Wallis method was used to conduct pairwise comparisons of image quality metrics across protocol settings.

The Kruskal–Wallis test, a non-parametric statistical method suitable for comparing three or more independent groups [31], was used to account for multiple comparisons, with results reported using a significance threshold of *p* < 0.005. This test assesses whether there is a statistically significant difference in the distribution of ranks among the groups.

## 3. Results

The mean quantitative image quality metrics of IQF_inv_ for all three chest protocol settings in both PA and LAT projection of the chest protocol is presented in the Table 2. An increase in image quality is positively correlated with higher IQF_inv_ values, suggesting that higher IQF_inv_ scores reflect improved image quality.

The comparison of physical grid, non-grid, and virtual grid protocols revealed clear differences in both image quality and radiation dose. The physical grid protocol consistently produced the highest quantitative image quality metric (IQF_inv_), with values increasing at higher INR levels (INR5 and INR8), but also resulted in the highest radiation exposure (mean DAP: 289.60 mGy·cm^2^ for PA, 286.60 mGy·cm^2^ for LAT). The non-grid protocol showed the lowest image quality but offered the most significant dose reduction (DAP: 46.60 mGy·cm^2^ for PA, 35.00 mGy·cm^2^ for LAT). The virtual grid demonstrated intermediate image quality while maintaining radiation dose levels comparable to the non-grid protocol, indicating a potential balance between image quality and dose optimisation. Across all protocols, image quality improved progressively from the non-INR (standard) setting to INR5 and INR8, reflecting the impact of increasing the image noise reduction (INR) software level. This enhancement was most pronounced in the physical grid protocol and less evident in the virtual grid and non-grid protocols.

### 3.1. Statistical Analysis of Image Quality for Chest Protocol Setting 1

The statistical pairwise comparison of the quantitative image quality metric IQF_inv_ across Non-INR, INR5, and INR8 for Chest Protocol 1 is presented in Table 3.

For chest protocol setting 1, both INR5 and INR8 significantly improve image quality (IQF_inv_) compared to non-INR for PA (*p* = 0.010, *p* < 0.001) and LAT (*p* = 0.006, *p* < 0.001) projections. INR8 consistently provides the highest image quality (QF_inv_), with a significant advantage over INR5 (PA: *p* = 0.015, LAT: *p* = 0.033), confirming its superior image quality.

Figure 3 presents boxplots comparing image quality metric (IQF_inv_) values across three INR settings in chest protocol 1, non-INR, INR5, and INR8, for PA and LAT projections. For the PA projection, in the non-INR setting, there is some variability, with an interquartile range (IQR) extending from approximately 2.50 to 3.00. In the INR5 setting, the IQR shows a range from about 3.25 to 3.75, indicating moderate variability. In the INR8 setting, the IQR extends from approximately 3.75 to 4.25, showing some variability but less than INR5. For the LAT projection, in the non-INR setting, variability is present, with an IQR extending from about 2.50 to 3.00. For the INR5 setting, the IQR shows a range from about 3.25 to 3.75, similar to the PA projection. In the INR8 setting for the LAT projection, the IQR extends from approximately 3.75 to 4.25, showing moderate variability.

### 3.2. Statistical Analysis of Image Quality for Chest Protocol Setting 2

To identify statistically significant differences between the various setups of chest protocol 2, pairwise comparisons of the image quality metric (IQF_inv_) between the INR and non-INR chest protocol 2 settings, using the Kruskal–Wallis test, are shown in Table 4.

In protocol 2, as shown in Table 4, the INR8 settings demonstrated significantly higher IQF_inv_ scores compared to INR5 in both projections. For the PA chest projection, INR8 scored significantly higher (*p* = 0.033), and for the LAT projection, INR8 also achieved a significantly higher score (*p* = 0.008). When comparing the INR and non-INR protocol settings, both INR5 (*p* = 0.006) and INR8 (*p* < 0.001) achieved significantly higher IQF_inv_ scores than the non-INR settings in the PA projection. Similarly, for the LAT projection, INR5 (*p* = 0.019) and INR8 (*p* < 0.001) also achieved significantly higher IQF_inv_ scores than the non-INR settings.

These statistical analyses revealed significant differences in quantitative image quality (IQF_inv_) across all protocol comparisons (non-INR vs. INR and INR5 vs. INR8) for both the PA and LAT projections. The non-INR protocols showed markedly lower performance compared to the INR protocols, and INR5 differed significantly from INR8, with INR8 achieving the highest IQF_inv_ scores overall.

The boxplot analysis comparing image quality between the INR-level chest protocol and the non-INR chest protocol and projections is presented in Figure 4.

### 3.3. Statistical Analysis of Image Quality for Chest Protocol Setting 3

The Kruskal–Wallis statistically significant comparison of the image quality for the INR levels and non-INR setting for chest protocol 3 and projection is presented in Table 5.

The Kruskal–Wallis analysis revealed statistically significant differences in the quantitative image quality metric IQF_inv_ between all protocol settings for both the PA and LAT projections (Table 5).

Boxplot comparisons of image quality among the non-INR, INR 5, and INR 8 settings of chest protocol 3 for both projections are shown in Figure 5.

As shown in the boxplot for chest protocol 3 in both the PA and LAT projections, INR8 consistently exhibits the highest IQF_inv_ scores, followed by INR5, with non-INR showing the lowest scores and the greatest variation in the PA projection. A similar trend is observed in the LAT projection, where INR8 achieves the highest IQF_inv_ scores, followed by INR5, and non-INR again shows the lowest scores. These results confirm that INR8 provides the highest image quality in both the PA and LAT projections, with INR5 demonstrating intermediate performance, while the non-INR protocols perform the lowest.

### 3.4. Statistical Analysis of Image Quality for Chest Protocol Setting 4

Table 6 presents the statistical comparison of DAP values across the grid settings for all three chest protocols and projections (PA and LAT). As shown in the table, there is no significant difference in DAP values between the virtual grid and non-grid (*p* = 0.380) for the PA projection. However, a significant difference in DAP values is found between the virtual grid and physical grid (*p* < 0.001) for the PA projection. Additionally, a significant difference is found between the non-grid and physical grid (*p* = 0.001) for the PA projection. For the LAT projection, there is also no significant difference in the DAP values between the virtual grid and non-grid (*p* = 0.254). However, a significant difference in DAP values was found between the virtual grid and physical grid (*p* < 0.001) for the LAT projection. Similarly, a significant difference was found between the non-grid and physical grid (*p* = 0.001) for the LAT projection.

Boxplots comparing DAP values across the three protocol settings, physical grid, non-grid, and virtual grid, for the PA and LAT chest projections are shown in Figure 6.

The boxplots indicate that the physical grid setting results in significantly higher DAP values compared to both the non-grid and virtual grid settings for both the PA and LAT projections. The non-grid and virtual grid settings show much lower DAP values with very little variability, suggesting they are more effective in reducing patient exposure to radiation while still maintaining image quality. The slight increase in DAP for the virtual grid compared to the non-grid suggests it might offer some advantages over the non-grid but at a slightly higher exposure cost.

### 3.5. Statistical Analysis of Image Quality for Chest Protocol Setting 5

The calculated ED in microsieverts (µSv) for the three protocol settings across the PA and LAT projections, along with percentage reductions relative to protocol 1 (physical grid), is summarised in Table 7. Protocol 1 yields the highest EDs (PA: 4.18 µSv; LAT: 3.89 µSv). Protocols 2 (non-grid) and 3 (virtual grid) significantly reduce the EDs: for the PA projection, the reductions are 76.8% and 77.0%, respectively, and for the LAT projection, both protocol settings achieve approximately 82% reductions.

## 4. Discussion

The novel INR software was evaluated and showed that significant differences in quantitative image quality and radiation dose were found between the protocols. Protocol 1, which used a physical anti-scatter grid, resulted in the highest radiation dose to the patient. In contrast, protocols 2 and 3 (without a physical anti-scatter grid) achieved a dose reduction of 77–82% for both the PA and LAT projections. This indicates a remarkably high physical anti-scatter grid factor in protocol 1. However, previous studies comparing physical and virtual anti-scatter grids have demonstrated a radiation dose reduction of 30–70% [13,14,15]. This higher anti-scatter grid contribution of radiation exposure may be attributed to the automatic exposure control (AEC) system, which appears to terminate exposure slightly prematurely in chest protocols without the physical anti-scatter grid.

A relatively recent study investigating the effect of INR on image quality, using multiple INR levels (0, 1, 3, 5, 7, and 10) in the pelvic protocol, demonstrated a correlation between higher INR levels and improved image quality (IQF_inv_) [9].

The initial INR levels ranged from 0 to 10. However, in this study, the image quality at INR levels 0, 5, and 8 was evaluated. The selection of these INR levels was based on the vendor’s recommendations and prior findings from a previously conducted study on visually assessed image quality, which indicated that the highest visual and quantitative image quality scores and the most clinically satisfactory results were obtained from the fifth INR level and above [9]. In the previously conducted study, INR levels were similarly assessed across various dose levels, all of which indicated an increase in image quality with increasing INR levels. The findings of the previous study are consistent with the results obtained using IQFinv in the present study.

The findings of the current study indicate that increasing the INR level leads to a decrease in image noise, thereby improving image quality. All three protocol settings demonstrated superior image quality at INR level 8 (INR8) compared to INR level 5 (INR5). The maximum INR level of eight was chosen based on the vendors’ recommendations for the current protocol.

The level setting of the virtual anti-scatter grid ranges from 1 to 10; however, in this study, level 5 at kernel 3 was used, as it corresponds to the default setting for the chest protocol as well as the recommendation from the vendor. The findings of this study have shown that the protocol incorporating a virtual anti-scatter grid demonstrated a 6% and 9% improvement in quantitative image quality (IQF_inv_) for the PA and LAT projections, respectively, compared to the protocol without an anti-scatter grid. The findings of this study indicate that the virtual anti-scatter grid can significantly enhance quantitative image quality for PA and LAT projections. Further investigation into the optimal level of virtual grid application may yield even greater improvements in image quality.

The chest protocol employing a physical anti-scatter grid achieved the highest qualitative image quality across all INR levels compared with protocols using no grid or a virtual grid; however, it entailed a four-to-six-time increase in patient radiation exposure. This study investigated the combined application of a virtual anti-scatter grid and INR software under the ALARA principle. The virtual grid attenuates scatter to reduce noise and radiation dose to the patients, while INR further augments noise reduction without additional exposure [13,14,15]. The results demonstrate that their integration yields synergistic improvements in IQF_inv_ at doses equivalent to or lower than those of either technique alone, thus delivering optimal image quality at the minimal radiation dose feasible for patients.

This study highlights the need for further research in optimisation of chest imaging protocols, particularly in light of the significantly higher radiation dose associated with protocol 1, which employs a physical anti-scatter grid, compared to protocols 2 and 3 that do not use a physical grid. Achieving an optimal balance between image quality and radiation exposure remains a critical goal. A potential strategy to address this involves integrating virtual grid technology with an appropriate INR level, which could facilitate the identification of a protocol adhering to the ALARA principle.

In this study, the image quality of a protocol incorporating INR levels and a virtual grid was quantitatively evaluated using a technical image quality phantom and compared to a conventional chest protocol using a physical anti-scatter grid. While this evaluation provides valuable insights, the results obtained should be validated through clinical assessments. Future studies should include visual image quality evaluations of these protocols by experienced radiologists or reporting radiographers to ensure the findings are robust and clinically relevant.

The use of technical phantoms to evaluate image quality and patient radiation dose has both advantages and limitations. One key limitation is that the evaluation is based on a phantom, which does not fully replicate a typical patient in terms of anatomy or body size. Specifically, the selected phantom size may not represent the full range of patient body types, thereby limiting the generalisability of the findings to the patient population. Furthermore, the phantom was used for both the PA and LAT projections, which do not accurately reflect the variation in patient thickness encountered in real clinical scenarios.

On the other hand, the use of technical phantoms offers advantages, such as the stability and uniformity of image quality assessment, as they are free from motion artefacts caused by patients or internal organ movement. Additionally, repeated image acquisitions could be performed without ethical concerns regarding radiation exposure, allowing for robust data collection.

## 5. Conclusions

This study highlighted the potential of INR technology to substantially enhance quantitative image quality across all three chest protocol settings. Chest protocol 1, incorporating a physical anti-scatter grid, demonstrated superior quantitative image quality compared to protocols using a virtual anti-scatter grid (chest protocol 3) or no grid (chest protocol 2). Furthermore, protocols integrated with INR software exhibited significantly higher quantitative image quality than those without INR integration. Higher INR levels were associated with better quantitative image quality, with INR level 8 acquisitions achieving significantly higher scores compared to INR level 5.

In addition to improved technical image quality, substantial dose reductions were achieved in chest protocols without a physical anti-scatter grid and protocol employing a virtual anti-scatter grid. These reductions ranged from 77% to 82% for both the PA and LAT projections, demonstrating the potential of these configurations for optimising both image quality and patient safety.

## Figures and Tables

**Figure 1 diagnostics-15-01391-f001:**
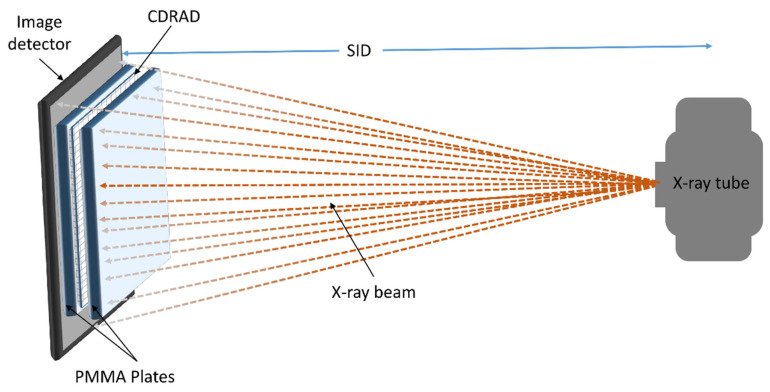
Schematic representation of the measurement setup using the CDRAD test object.

**Figure 2 diagnostics-15-01391-f002:**
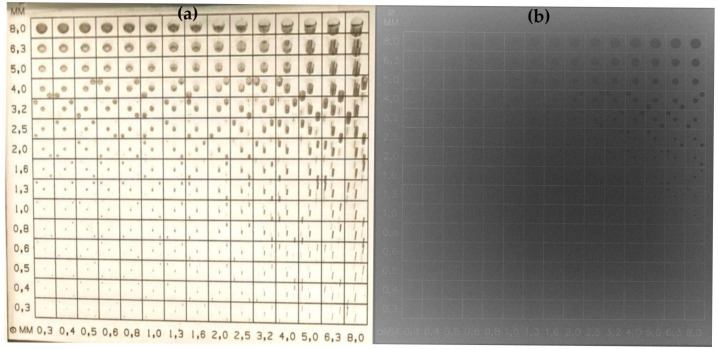
Photographic image (**a**) and radiographic image (**b**) of the CDRAD phantom used for quantitative image quality assessment.

**Figure 3 diagnostics-15-01391-f003:**
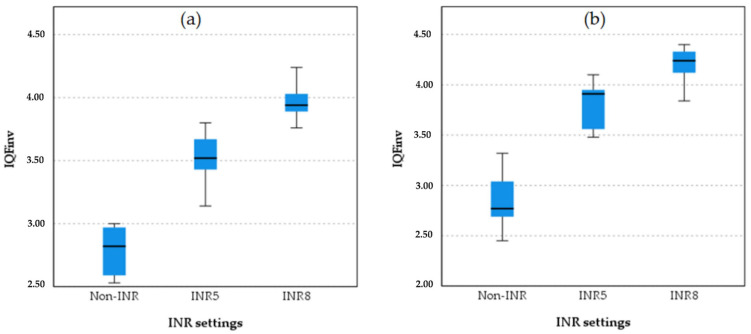
Boxplot comparison quantitative image quality metric (IQF_inv_) across INR settings (non-INR, INR5, and INR8) for the PA projection (**a**) and LAT projection (**b**).

**Figure 4 diagnostics-15-01391-f004:**
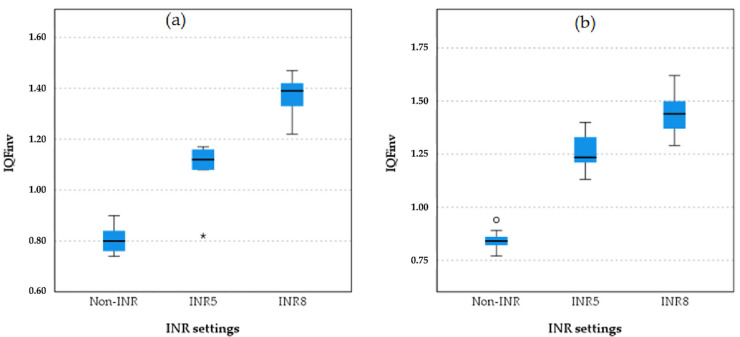
Boxplot comparison of quantitative image quality between different INR levels and non-INR setting for both PA and LAT projection in chest protocol 2 (without anti-scatter grid). The boxplots illustrate the IQF_inv_ scores for different protocol settings in the (**a**) PA and (**b**) LAT projections in protocol 3. INR8 consistently demonstrates the highest IQF_inv_ scores in both projections, followed by INR5. Non-INR settings have the lowest IQF_inv_ scores, with clear separation between the non-INR and INR protocols. This pattern reflects the superior quantitative image quality of INR8 across both projections. The star symbol (*) indicates that one of the IQFinv values for INR5 was lower than most others and falls beyond the whiskers of the box plot.

**Figure 5 diagnostics-15-01391-f005:**
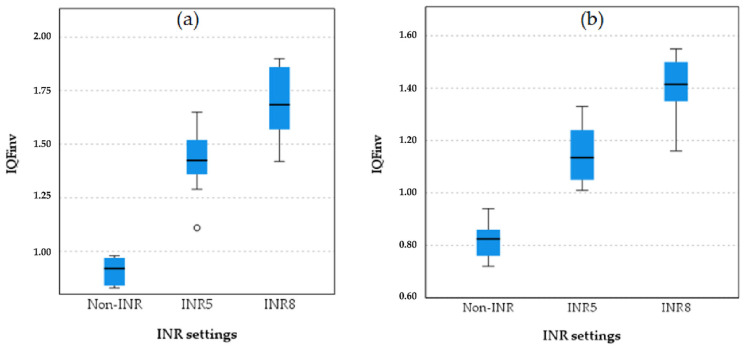
Boxplot comparison of quantitative image quality metric (IQF_inv_) between different INR levels and non-INR setting for both PA (**a**) and LAT (**b**) projection in chest protocol 3 (with virtual anti-scatter grid). The small circle in plot (**a**) represents an outlier a data point that lies outside the expected range of values in the INR5 group.

**Figure 6 diagnostics-15-01391-f006:**
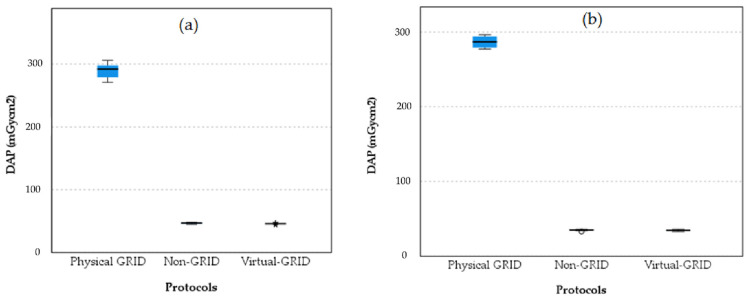
Boxplot comparison of DAP values across three protocol settings for PA projection (**a**) and for LAT projection (**b**). The barely visible boxplots for Non-GRID and Virtual-GRID show significantly lower DAP values than the Physical GRID, which indicates a lower radiation exposure.

**Table 1 diagnostics-15-01391-t001:** Chest protocol settings at different INR levels, including geometric and exposure parameters, and anti-scatter grid configurations. SID = source-to-image distance; Cu = Copper, PA = posterior–anterior projection; LAT = lateral projection.

Protocols	INR Levels	Grid Settings	Projections	Tube Energy (kV)	Tube Filtration (Cu)	SID (cm)	Focus Size	Exposure Type
Protocol 1	Non-INR	Physical grid	PA/LAT	125	0	180	Large	AEC
	INR 5							
	INR 8							
Protocol 2	Non-INR	Non-grid	PA/LAT	125	0	180	Large	AEC
	INR 5							
	INR 8							
Protocol 3	Non-INR	Virtual grid	PA/LAT	125	0	180	Large	AEC
	INR 5							
	INR 8							

**Table 2 diagnostics-15-01391-t002:** Results of image quality metrics (IQF_inv_) and DAP values for all three chest protocol settings and projections (PA and LAT) across different INR levels.

Protocols	Settings	Mean IQF_inv_	Mean DAP (mGycm^2^)	Projections
Protocol 1	Physical Grid	2.79	289.60	PA
	INR 5	3.57		
	INR 8	4.02		
	Physical Grid	2.74	286.60	LAT
	INR5	3.81		
	INR8	4.25		
Protocol 2	Non_GRID	0.88	46.60	PA
	INR5	1.28		
	INR8	1.47		
	Non_GRID	0.83	35.00	LAT
	INR5	1.15		
	INR8	1.35		
Protocol 3	Virtual_GRID	0.85	45.8	PA
	INR5	1.21		
	INR8	1.45		
	Virtual_GRID	0.91	34.5	LAT
	INR5	1.40		
	INR8	1.60		

**Table 3 diagnostics-15-01391-t003:** Statistically significant comparison of the quantitative image quality (IQF_inv_) between the INR and non-INR for chest protocol setting 1 and PA and LAT projections. Bold formatting of the *p*-values indicates significant differences across the INR settings.

Protocol Comparison	Test Statistic	Std. Error	Std. Test Statistic	*p*-Values	Projections
Non-INR vs. INR5	−10.200	3.936	−2.591	**0.010**	PA
Non-INR vs. INR8	−19.800	3.936	−5.030	**<0.001**	
INR5 vs. INR8	−9.600	3.936	−2.439	**0.015**	
Non-INR vs. INR5	−10.800	3.937	−2.744	**0.006**	LAT
Non-INR vs. INR8	−19.200	3.937	−4.877	**<0.001**	
INR5 vs. INR8	−8.400	3.937	−2.134	**0.033**	

**Table 4 diagnostics-15-01391-t004:** Statistical pairwise comparison of quantitative image quality (IQF_inv_) between non-INR and INR settings for chest protocol 2 in the PA and LAT projections. Bold formatting of the *p*-values indicates significant differences across the INR settings.

Protocol Comparison	Test Statistic	Std. Error	Std. Test Statistic	*p*-Values	Projections
Non-INR vs. INR5	−10.80	3.94	−2.74	**0.006**	PA
Non-INR vs. INR8	−19.20	3.94	−4.88	**<0.001**	
INR5 vs. INR8	−8.40	3.94	−2.14	**0.033**	
Non-INR vs. INR5	−9.20	3.93	−2.34	**0.019**	LAT
Non-INR vs. INR8	−19.60	3.93	−4.98	**<0.001**	
INR5 vs. INR8	−10.40	3.93	−2.64	**0.008**	

**Table 5 diagnostics-15-01391-t005:** Statistically significant pairwise comparison of the quantitative image quality metric (IQF_inv_) between the INR and on-INR setting for chest protocol 3 (with virtual anti-scatter grid) and projections.

Protocols Comparison	Test Statistic	Std. Error	Std. Test Statistic	*p*-Value	Projections
Non-INR vs. INR5	−10.550	3.937	−2.68	**0.007**	PA
Non-INR vs. INR8	−19.450	3.937	−4.94	**<0.001**	
INR5 vs. INR8	−8.900	3.937	−2.26	**0.024**	
Non-INR vs. INR5	−11.000	3.936	−2.80	**0.005**	LAT
Non-INR vs. INR8	−19.000	3.936	−4.83	**<0.001**	
INR5 vs. INR8	−8.00	3.94	−2.03	**0.042**	

**Table 6 diagnostics-15-01391-t006:** Statistical pairwise comparison of DAP values for chest protocols with physical anti-scatter grid, virtual anti-scatter grid, and on-grid settings and projections.

Protocol Comparison	Test Statistic	Std. Error	Std. Test Statistic	*p*-Values	Projections
Virtual grid vs. non-grid	3.40	3.87	0.879	0.380	PA
Virtual grid vs. physical grid	16.70	3.87	4.315	**<0.001**	
Non-grid vs. physical grid	13.30	3.87	3.437	**0.001**	
Virtual grid vs. non-grid	4.40	3.86	1.140	0.254	LAT
Virtual grid vs. physical grid	17.20	3.86	4.457	**<0.001**	
Non-grid vs. physical grid	12.80	3.86	3.317	**0.001**	

**Table 7 diagnostics-15-01391-t007:** ED and percentage differences between physical grid, non-grid, and virtual grid settings for chest protocol projections (PA and LAT).

Protocols	ED (µSv)	Projections	Difference (%)
Protocol 1	4.18	PA	
	3.89	LAT	
Protocol 2	0.97	PA	76.8
	0.68	LAT	82.5
Protocol 3	0.96	PA	77.0
	0.71	LAT	81.7

## Data Availability

The data collected for this study were not made publicly available. Phantom images, dose assessments, associated analyses and calculations, as well as the manuscript, were stored on a local personal computer (PC) hard drive.

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
