# Peer review of "Quantitative Evaluation of Image Quality and Radiation Dose Using Novel Intelligent Noise Reduction (INR) Software in Chest Radiography: A Phantom Study"

_diagnostics, 2025, doi:10.3390/diagnostics15111391_

Round 1

Reviewer 1 Report

Comments and Suggestions for Authors
  1. Does the manuscript explicitly define the research goals and hypotheses regarding INR software across different chest radiography protocols?
  2. Does the phantom setup (CDRAD 2.0 with 20 cm PMMA) accurately simulate clinical chest radiography?
  3. Are the INR level selections (0, 5, and 8) well justified based on previous literature or vendor recommendations?
  4. Would additional detail improve reproducibility?
  5. Was the use of the Kruskal-Wallis test justified for comparing IQFinv values?
  6. Are post-hoc comparisons and p-values interpreted correctly?
  7. Are Tables 2–7 and Figures 1–5 clear and informative?
  8. Is the distinction between PA and LAT projections sufficiently highlighted?
  9. Are the claimed benefits of INR8 over INR5 and non-INR settings statistically and clinically significant?
  10. Are the 77–82% reductions realistic, and do they align with previously published data?
  11. Is the combination of INR software with a virtual anti-scatter grid a novel contribution to radiographic dose optimization?
  12. Are there realistic implementation pathways or limitations outlined for clinical adoption?
  13. Does the manuscript adequately acknowledge the limitations of phantom-based assessments?
  14. Are recommendations for clinical validation and visual image quality studies appropriately emphasized?
  15. Are there any grammar, syntax, or formatting issues that need correction?
  16. Is the use of terminology (e.g., IQFinv, DAP, ED) consistent throughout the paper?
  17. Need to the figures and tables include appropriate captions and units?
  18. Are the technical and clinical white papers cited for Canon INR software publicly accessible?

Author Response

The response to the reviewer is attached, and the revised parts of the main text are marked in blue.

Reviewer 2 Report

Comments and Suggestions for Authors

This is a paper that applies an INR program for reducing image noise and radiation dose, and is applicable in clinical practice. This is a paper that can contribute to radiation dose reduction by applying it to patients through additional research.

  1. In results. There are no images of chest X-ray examinations using the phantom in the paper. Additional relevant experimental images need to be added to the results.
  2. In effective dose estimation. In comparing doses according to protocols, it is necessary to additionally mention the grid ratio according to grid use.
  3. Line 246. The manuscript needs a detailed explanation of Physical Grid, Non-Grid, and Virtual Grid.
  4.  In discussion. Given the limitations of the study, additional experiments are warranted to compare images of patients with a control group based on the results of the phantom experiments.

Thank you.

Author Response

(The authors gave the same response as above.)

Reviewer 3 Report

Comments and Suggestions for Authors

Authors in this study tested three chest protocols comprising protocol 1 (physical anti-scatter grid), protocol 2 (non grid) and protocol 3 (virtual anti-scatter grid) with acquisition of images based on three levels, namely non-INR, INR-5 and INR-8. Their results showed significant improvement of image quality with use of novel INR technology. Significant dose reduction was found in protocols 2 and 3 when compared to protocol 1 with superior image quality. The manuscript is nicely written and easy to follow the contents. While the study presents interesting findings, it will need a revision before it can be accepted for publication. My main concerns are about lack of some description of the methods as it is necessary to provide details for readers to understand better if they are not very familiar with the INR technology. 

Specific comments:

  1. Abstract:  this sentence : "Future clinical validation, incorporating 
    subjective visual assessments by radiologists, is recommended to confirm these findings and facilitate the integration of INR closer to clinical practice" does not belong to the results. Suggest moving it to Conclusion.
  2. Introduction: good literature with study purpose clearly stated. 
  3. Methods: IQF abbreviation was provided but it was repeated many times, so suggest just using the IQF once abbreviation was provided. Further, I am not very familiar with the IQF as usually we use SNR and CNR for quantitative assessment of image quality. I would suggest that authors provide further clarification of the use of IQF and how it compares to the standard SNR/CNR. 
  4. Methods-statistical analysis: is Kruskal-Wallis sufficient for comparing the image quality from different protocols? Please double check it with a biostatistician. 
  5. Results are well supported by tables and figures. It would be nice to provide one or two figures showing the phantom images acquired from these protocols. 
  6. Discussion: most of the contents in your discussion are more like presentation of results while there is lack of detailed discussion of what your findings mean, in particular how your findings compared to other studies. Please enhance it by comparing your study with others available in the literature. 

Author Response

(The authors gave the same response as above.)

Round 2

Reviewer 1 Report

Comments and Suggestions for Authors

Nil

Reviewer 3 Report

Comments and Suggestions for Authors

Thank you for addressing my comments in the revised manuscript which shows improvements compared to the original submission.